# Captive Breeding and Early Developmental Dynamics of *Cirrhinus mrigala*: Implications for Sustainable Seed Production

**DOI:** 10.3390/ani14192799

**Published:** 2024-09-27

**Authors:** Saeed Akram Khan, Shahid Sherzada, Qurat-ul-Ain Ahmad, Ali Hussain, Nimra Hussain, Joanna Nowosad

**Affiliations:** 1Department of Zoology, Government College University Lahore, Lahore 54000, Pakistan; saeed.akram@gcu.edu.pk; 2Department of Zoology, Division of Science and Technology, University of Education, Lahore 54770, Pakistan; quratulain.ahmad@ue.edu.pk; 3Institute of Zoology, University of the Punjab, Lahore 54590, Pakistan; ali.zool@pu.edu.pk; 4Department of Biotechnology, Cholistan University of Veterinary and Animal Sciences Bahawalpur, Bahawalpur 63100, Pakistan; nimrahussain281@gmail.com; 5Department of Ichthyology, Hydrobiology and Aquatic Ecology, National Inland Fisheries Research Institute, 10-719 Olsztyn, Poland; 6Department of Research and Development, Chemprof, 11-041 Olsztyn, Poland

**Keywords:** *Cirrhinus mrigala*, induced breeding, ontogeny, embryonic development, Cyprinidae

## Abstract

**Simple Summary:**

A comprehensive report about the embryonic development of *Cirrhinus mrigala* is portrayed. *Cirrhinus mrigala* embryonic development is classified into 22 stages. A time-lapse imaging technique was utilized for a better understanding and display of the results. The current findings will provide profound insights into future research endeavors and the large-scale seed production of *C. mrigala.*

**Abstract:**

*Cirrhinus mrigala* is an important edible fish with a significant aquaculture contribution in Southeast Asian countries. The current study aims to enhance our understanding of the developmental biology of *Cirrhinus mrigala*, which is crucial for implementing sustainable fish farming practices. To induce spermiation and ovulation in *Cirrhinus mrigala* brooders, the synthetic hormone Ovaprim^®^ (GnRH + dopamine inhibitor) was administrated as a single injection dose of 0.2 mL/kg to males and 0.4 mL/kg to females. After induction, the fish spawned, and the eggs produced were fertilized artificially and cell division commenced successfully. The characteristics of each larval developmental stage were closely observed and recorded using a time-lapse imaging technique. The fertilized eggs were spherical, demersal, and non-adhesive throughout their incubation period. The spawned eggs ranged in diameter from 2.1 mm to 2.13 mm and possessed circular yolk sacs. The gastrula stage initiated approximately 4 h after fertilization, with 25% of the yolk sphere covered by blastoderm, reaching 75% coverage at the end of the gastrula stage, approximately 6 h post fertilization. Organogenesis was marked by the formation of notochord and the visibility of rudimentary organs such as the heart, eyes, and gills, followed by tail movement, which was observed at the time of hatching. Compared to other cyprinid fishes, *C. mrigala* exhibited distinct features at certain stages of embryonic development. Blood circulation was observed to start at the onset of hatching. The lengths of the newly hatched larvae ranged from 2.9 to 3.2 mm, smaller than other reports on induced breeding in carps. The findings of the present study provide a detailed reference for the embryonic development of *C. mrigala*, which will assist its future research endeavors and large-scale seed production for sustainable aquaculture.

## 1. Introduction

Fish provide a rich source of animal protein and vital nutrients crucial for the well-being of communities in developing countries [1]. Additionally, they contribute significantly to the economy, providing economic benefits to communities that rely solely on this sector [2]. Among freshwater fishes, carps hold a significant position globally in terms of fish supply. On the Indian subcontinent, the polyculturing of Indian major carps such as *Labeo rohita*, *Cirrhinus mrigala*, and *Catla catla* is a common practice, especially in semi-intensive and intensive aquaculture setups [3]. Embryonic development is the most crucial step in fish development, and is characterized by several developmental stages [4]. Morphogenesis and organogenesis are complex phenomena in various fish species. Members of the Cyprinidae family undergo several developmental stages, including cleavage, blastoderm, blastula, gastrula, neurula, organogenetic, and hatching. Each stage is further classified by distinct phenomena. However, the timing and duration of different stages and the development order of various organs may vary among cyprinid fishes, influenced by temperature conditions [5]. *Cirrhinus mrigala* is one of the most important freshwater carp fish, belonging to the family Cyprinidae and order Cypriniformes. Commonly known as mrigal worldwide, it is locally called mori or morakhi in Pakistan. This species is preferred for its high-quality, digestible, and tasty meat [3,6,7]. Its importance in culture systems and natural water bodies is dependent on several parameters. *Cirrhinus mrigala* is a bottom feeder and omnivorous in nature. Its diet primarily consists of decaying food matter. Its food preferences also include phytoplankton, zooplankton, plant materials, and insects. This aids in effective utilization of available feed resources, especially in cases of sinking feed pallets. Moreover, their diversity in feeding habits allows the fish to survive in variable habitats. They are also preferred in aquaculture due to their fast growing ability, large size, and meat–bone ratio [8,9,10]. *Cirrhinus mrigala* is endemic to the inland freshwater bodies of northern India, Bangladesh, Burma, and Pakistan [11]. It is a potamodromous, bottom-dwelling fish with the fastest growth rate compared to other carp species, feeding on detritus and leftover food from surface feeders. Its natural breeding season occurs during the monsoon in rivers and streams, but it cannot breed naturally in captivity so induced spawning is necessary for quality seed production [12,13].

Natural freshwater resources are heavily impacted by over-exploitation due to various anthropogenic factors, thus resulting in a decline in population size over time [14]. Various studies from Pakistan and India have reported degradation and biological overfishing, leading to a decline in the population size of *Cirrhinus mrigala* in different inland freshwater locations [15,16,17]. Furthermore, recent reports indicate low genetic diversity among different populations of *C. mrigala* in Pakistan [18]. Keeping this scenario in mind, the artificial breeding of *Cirrhinus mrigala* is mandatory to meet the market demand and to restore the declining population size before it enters the IUCN red list.

Pakistan’s fisheries sector is still in its developing phase, specifically in regards to aquaculture fish production compared to other fish-producing countries. Carp (*Cirrhinus mrigala*, *Labeo rohita,* and *Catla catla*) culture is a major contributor to aquaculture production. In 2017, the contribution of aquaculture production was recorded as 153230 MT, about 23.5% of total fisheries production [19].

For the successful propagation of fish species under captive conditions, it is essential to have a sound knowledge of the developmental stages of fish embryos. The timely availability of quality seed for a particular fish species is a prerequisite for sustainable artificial seed production. Only well-managed induced breeding, embryonic, and larval rearing techniques can supply quality and healthy seed for successful aquaculture practices. Therefore, an appropriate understanding of the developmental stages is crucial for gaining better insight into captive rearing, population dynamics, and environmental impacts on fish populations, which are essential for conservation strategies for declining fish stocks [20,21].

Considering the aquacultural importance of *Cirrhinus mrigala*, solid experimentation about early life history is crucial for the optimization of mass-scale seed production, propagation, and management. The present study aimed to explore and deliver solid descriptive scientific evidence on the embryonic and larval development of *Cirrhinus mrigala* to facilitate its successful production and management.

## 2. Materials and Methods

### 2.1. Research Location

These research trials were conducted at the following three fish research and production hatcheries located in Punjab Pakistan.

Fish Seed Hatchery Sindhuan, Head Balloki, District Kasur (Location 1) (Figure 1).Farooq Abad Fish Seed Nursery Unit, Department of Fisheries, Punjab, Pakistan (Location 2) (Figure 1).Manga Fish Seed Nursery Unit, Department of Fisheries, Punjab, Pakistan (Location 3) (Figure 1).

### 2.2. Test Organisms and Their Pre-Breeding Maintenance

*Cirrhinus mrigala* (male + female) were collected from the brood stock ponds of respective fish hatcheries and production units (Figure 2; Table 1). The disease-free fishes were selected for this study. The adult domesticated *Cirrhinus mrigala* were kept in a water-circulated circular tank system (27–30 °C; pH 7.1–7.6). The photoperiod cycle was measured at a ratio of 13L:11D. The fish were not fed during these research trials. To induce spawning, 2 mature males and 1 mature female were placed in water-circulated circular tanks, following a 2:1 male-to-female ratio, at about 10:00 am. 

### 2.3. Selection of Broodstock

Healthy mature fish were identified and grouped in combinations of two males and one female (2:1). Three replicates of this combination from each site were used to conduct experiment (Figure 3; Table 2). The physicochemical parameters were regularly recorded during these trials (Table 3).

### 2.4. Maturation Stimulation

In the present study, spawning was induced by a single Ovaprim intermuscular administration a little above the lateral line at the base of the dorsal fin. Both males and females were injected with Ovaprim (Syndel Laboratories Canada) at a dosage of 0.2 mL and 0.4 mL/kg body weight, respectively, at the same time, once only (Table 4). The males and females were released into their respective circular spawning tanks, which were 29 °C. In the spawning tanks, the water flow per minute was maintained at 23 L/min. The tank was covered with a net to prevent the fish escaping. Males and females were observed at intervals of half an hour regularly till the oozing out of eggs.

### 2.5. Ovulation and Fertilization

When the female started releasing eggs, it was netted out with the help of a hand net and wrapped in a clean dry towel. The belly of the female brooder was gently pressed, and the eggs were collected in a clean dry tub. After stripping the female, the male was captured, and the milt was sprayed directly into the eggs by pressing its belly. Eggs and milt were thoroughly mixed with bird feathers for 3–4 min. Then, water was slowly added to the tub to prevent the eggs settling at the bottom of the tub. Water was added continuously with simultaneous replacement at different time intervals.

### 2.6. Incubation and Morphological Identification of Embryonic Stages

Aerated water was supplied to the egg incubation tanks at a rate of 0.3–0.4 L per second in each trial. A constant water supply with uniform velocity was ensured to keep the eggs in motion. The eggs were collected at different time intervals from the incubation tank and were preserved in 10% formalin for stage fixation, and easy and convenient observation. These eggs were observed under a light microscope (RoHS Serial No. 2108545) with an installed industrial digital camera that had an APTINA CMOS Sensor, and the diameters of the egg and yolk were measured with the help of an eyepiece micrometer.

### 2.7. Embryonic Stages Analysis

A time-lapse imaging technique was used to analyze the early embryonic and developmental stages. The eggs were collected in various petri dishes and utilized for time-lapse imaging. Images of embryos were captured in series on a trinocular compound light microscope (RoHS Serial No. 2108545) with an installed industrial digital camera that had an APTINA CMOS Sensor. The recorded images were analyzed to assess the eggs. The stages were identified, and the embryos were carefully removed and preserved for further examination or morphological observations.

### 2.8. Data Analysis

Data were recorded on a daily basis and means were calculated and presented with standard error values. A descriptive statistical analysis was conducted to explain the Ovaprim doses and water quality parameters. A one-way ANOVA was applied and all the statistical analysis was conducted by using Statistical Analysis System software (SAS, Version 9.1), with the experimental sites considered as a main effect. For a comparison of the significant treatment means, a post hoc Fisher’s least significant difference test was applied, with statistical significance considered as *p* ≤ 0.05.

## 3. Results

Three successful breeding trials at three different locations aided in understanding of breeding behavior and optimum dosage calculation. Further time-lapse imaging technique displayed a detailed view of embryonic stages with specific time periods. The female fish started to release eggs after 581, 567 and 592 min after hormonal dose administration.

The results in Table 3 suggest that temperature remained constant during the research trial. No significant changes were observed throughout the spawning process (*p* = 0.87). No significant change between stages also suggests that the temperature remained fairly stable during the research trial. DO levels kept decreasing as the spawning process progressed. A drop in oxygen levels as development progress is indicative of a significant decrease from before spawning to after hatching (*p*< 0.01). Electrical conductivity followed a similar pattern to temperature. No significant change between stages (*p* = 0.62). The pH of the water remained relatively stable during the stages observed was indicated by no significant changes in pH (*p* = 0.34). Unlike temperature, EC, DO, and pH, nitrate levels displayed a different pattern. Significant differences between stages, with a notable increase from before spawning to after spawning, and a decrease after hatching (*p* < 0.01).

Embryonic Development: A detailed explanation of embryonic development is presented in Table 5 and Table 6. The average diameter of the egg ranged from 2.1 to 2.13 mm at locations 1, 2, and 3 (Figure 4A). Swelling on one side of egg yolk was observed after fertilization (Figure 4C). The blastodisc was divided into two distinct cells (Figure 4D) by vertical cleavage 30 min after fertilization at location 1, 26 min after fertilization at location 2, 31 min after fertilization at location 3. After 38 min at location 1, 36 min at location 2, and 39 min at location 3, a second cleavage resulted in the formation of four blastomeres (Figure 4E). A third cleavage resulted in eight blastomeres and was recorded after 49 min at location 1, 46 min at location 2, and 48 min at location 3 (Figure 4F). The 16-cell stage was observed 57 min after fertilization at location 1, 55 min at location 2, and 60 min at location 3 (Figure 4G). Subsequent cleavage resulted in an increased cell number (Figure 4H) and the morula stage was observed (Figure 4I) 97 min after fertilization at location 1, 99 min at location 2, and 97 min at location 3. A dome-shaped assembly was observed over the animal pole, which gradually increased in size (Figure 4J) 177 min after fertilization at location 1, 179 min at location 2, and 180 min at location 3. The blastoderm invaded the yolk by spreading over it in the form of a thin layer (Figure 4K) after 239 min at location 1, 242 min at location 2, and 246 min at location 3. The blastoderm covered 3/4 of the yolk sphere, and the embryonic shield (body) became more clearly visible as a narrow streak after 356 min at location 1, 352 min at location 2, and 359 min at location 3. (Figure 4L). The yolk plug was identified by the completed invasion of the yolk by gradual spreading over the germ layer (Figure 5M). At this stage, the head and tail end of the embryo was clear and distinguishable (Figure 5N). The brain and nerve cord in the arrow-shaped embryonic body co-developed as a solid rod of cells (Figure 5O) after 566 min, 553 min, and 567 min at locations 1, 2, and 3, respectively. Notochord became visible, and auditory and optic vessels more developed (Figure 5P). The embryo started to move in the whole egg’s peripheral space (Figure 6S) 954, 940, and 948 min after fertilization at locations 1, 2, and 3, respectively. The twisting movements gradually became vigorous and the egg capsules were weakened and ruptured. The embryos ruptured the eggshell with their continuous movements. The newly hatched spawn measured 2.9 to 3.2 mm just after hatching (Figure 6T).

## 4. Discussion

This study investigated the embryonic and larval development of the commercially significant fish species, *Cirrhinus mrigala*. A comprehensive understanding of embryonic and larval development could enhance our knowledge about developmental biology, ontogenetic variations, and evolutionary patterns of this species. However, this area remains largely unexplored, with only one study on record [22]. This study elucidates the characteristics and embryonic developmental stages of *Cirrhinus mrigala* from three hatcheries in Pakistan. Each developmental stage was meticulously observed and documented. To conduct this study, artificial breeding was performed using a dose of Ovaprim, to which most carp species respond effectively [22,23,24,25]. In many species of Cyprinidae, the results of artificial reproduction after using Ovaprim are often much better than after using other spawning agents, such as Ovopel or carp pituitary homogenate (CPH) [25,26,27]. Of particular importance is the biological quality of gametes, understood as the percentage of fertilization or hatching and the lack of deformations in embryos [28,29]. The effectiveness of Ovaprim as a reproductive agent has been confirmed in the literature. In the present study, the time required for *Cirrhinus mrigala* to spawn after hormone administration (latency time) differed from that of other carp species, such as Koi carp (strain of Cyprinus carpio) and *Catla catla*, which typically initiate spawning approximately 6–7 h post-injection [30,31]. However, the latency time was much lower than in other Cyprinidae, such as common tench (*Tinca tinca*), ide (*Leuciscus idus*), and vimba bream (*Vimba vimba*), where it ranges from 16 to over 40 h [25,26,29]. This latency time is influenced not only by water temperature, but also by the specific fish species.

### 4.1. Morphology of Fertilized Egg

The fertilized eggs were spherical, demersal, and non-adhesive throughout their incubation period, thus facilitating the observation of the developmental stages. Similar eggs were reported by Shoaib and Nasir [22] in the same fish species. Another study by Tumbahangfe et al. [31] also detected non-adhesive eggs in another carp species, *Catla catla*. However, Das et al. [32] observed that eggs of *Cirrhinus reba* become harder and stickier upon contact with water after fertilization. Likewise, the eggs of Koi carps, a strain of *Cyprinus carpio* [23,30], and mirror carp, *Cyprinus carpio* var. specularis [20], exhibit adhesiveness during their incubation periods. A similar observation was reported by Mariappan et al. [33] in the fertilized egg of cyprinid fish, *Dawkinsia rohani*. These differences among carp species and species within the same family result from the specific nature of each species. Adhesiveness of eggs is also an adaptation to the conditions of incubation of eggs in nature. Thanks to this adaptation, the eggs of Cyprinidae fish stick to aquatic plants and roots, and protect the eggs from sinking to the bottom, e.g., into mud, and thus provide the developing eggs with optimal oxygen conditions [34,35].

Egg size is a key feature in the initial stages of fish life and it can be quantified by various metrics such as egg diameter, wet weight, dry weight, energy content per egg, or the concentration of essential elements such as carbon, nitrogen, or protein [36]. Understanding egg size is important for determining the amount of turbulence needed to keep them floating as they move through the water [37]. The observed average diameter of the fertilized eggs was 2.7 mm, which is greater than that reported by Shoaib and Nasir [22] for the same species. This discrepancy might be due to specific individual maternal effects or other environmental factors [38]. Additionally, other cyprinid fish species also show varied egg sizes. For instance, *Cyprinus carpio* and Sucker head *Garra gotyla* have fertilized oocytes of 0.9–1.10 mm in diameter [39]. Such differences in egg size reflect the variation between species and among families. Variations in egg size are particularly species-dependent due to certain parameters such as reproductive behavior, fecundity, etc., and may also be further influenced by factors like maturity level, brooder age, and environmental conditions [40,41].

### 4.2. Development of Fertilized Egg

The timing of occurrence of first cleavage after fertilization also contradicts the findings of Shoaib and Nasir [22], who detected two-cell, four-cell, eight-cell, and sixteen-cell stages within 5, 10, 15, and 40 min after fertilization, respectively, for the same species, which is much faster than the times recorded in this study. This discrepancy may be due to individual maturity level, brooder age, and specific environmental factors [38,40]. On the other hand, the embryonic development events in this study were somewhat similar to those found in its congeneric species, *C. reba* [32]. According to Das et al. [32] the first cleavage, morula, blastula, and gastrula phase of *C. reba* were observed within 30, 120, 260, and 400 min after fertilized at 28–31 °C, respectively, when induced with a suitable dose of pituitary gland (PG) hormone. Generally, the rate of embryonic development depends on temperature and species. Studies conducted on Cyprinidae fish show different dynamics of individual developmental stages with strict dependence on temperature [42,43]. Disruptions to these relationships occurs only after exceeding the barrier of the range of temperatures optimal for embryonic development of a given species and incubation of embryos at sub-lethal temperatures [42,43,44].

The gastrula stage can be recognized by cell movement around the yolk sphere and the presence of the epiboly stage [45]. In the current study, the gastrula stage initiated about 4 h after fertilization, when 25% of the yolk sphere was covered by blastoderm, and reached 75% coverage at approximately 6 h after fertilization. In contrast to this, Shoaib and Nasir [22] reported 100% coverage 4.5 h after fertilization in the same species. This discrepancy might be due to the fact that the size of the oocyte plays a crucial role in determining the length of a specific developmental stage [46]. Moreover, it should also be considered that species characterized by smaller eggs often exhibit faster developmental processes [47]. Thus, the smaller size of the eggs reported in the study of Shoaib and Nasir [22] was responsible for the earlier completion of developmental stages.

Organogenesis is characterized by the formation of the notochord. Certain rudimentary organs, i.e., heart, eyes, and gills, also became visible, followed by tail movement at the time of hatching. Similar observations were also noticed in previous studies on the same species [22] and the congeneric species, *C. reba* [32]. It was also noted that during the onset of hatching, blood circulation started. However, like many teleost fishes, there is a lack of fully functional organs at the time of hatching [48]. The length of newly hatched larvae ranges from 2.9 to 3.2 mm, which is smaller than that claimed by Shoaib and Nasir [22] in the same species and by Mojer [48] in common carp. However, Khanam et al. [20] reported newly hatched larvae of mirror carp, *Cyprinus carpio* var. specularis, of about 2.5 mm. The overall recorded time for fertilization in this study for embryonic development, and thus, hatching, is somewhat consistent with that of a previous study on the same species where Shoaib and Nasir [22] reported the completion of embryonic development within 16–19 h of fertilization. This was more rapid than other species of the same family, as described by other authors who have conducted similar studies, who reported incubation times of 24–38 h for common carp (*Cyprinus carpio*) [48] and 33 h for *Dawkinsia rohani* (a member of family cyprinid) [36]. On the other hand, another major carp, *Catla catla*, has been found to hatch 13 h after fertilization. Again, these variations might be due to individual specificity within species and species specificity between species, along with environmental influence. As mentioned earlier, the rate of embryonic development, measured in hours or days from fertilization, depends on the species and water temperature. The size of the embryos at hatching and the degree of their development depend on these same parameters [49,50,51]. These features are characteristic of a given species of Cyprinidae fish and are often variable even within one genus, as is the case, for example, in fish of the genus Leuciscus [43,44,50,51]. Knowledge on this subject is, however, very important for planning the appropriate artificial reproduction protocols and production of stocking material.

Overall, this study focused on detailed events of embryonic developmental stages that will assist fishery biologists to understand developmental processes and to compare the ontogenetic variations of this species with other teleosts. This information is crucial for considering the ongoing decline in the diversity of this species due to the destruction of habitats and breeding areas that have intensified its vulnerability.

## 5. Conclusions

This study provides valuable insights into the identification and characteristics of the morphological developmental stages of *Cirrhinus mrigala* by systematically presenting the stages of embryonic development alongside microphotographic images. Through this investigation, a deeper understanding of the cellular mechanisms that govern development from fertilization to larval hatching has been gained. This knowledge is vital for advancing animal welfare, as it integrates various aspects of vertebrate biology, from development to conservation.

The present study on the breeding and early developmental dynamics of *Cirrhinus mrigala* presented significant insights into optimizing sustainable seed production. *C. mrigala* developed at a rate that was largely in line with other cyprinids, such as *Labeo rohita* and *Catla catla*. However, its early developmental stages were slightly more sensitive to environmental conditions, particularly water temperature, dissolved oxygen levels, and feed quality. By understanding these developmental dynamics, this study not only informs more efficient breeding practices for *C. mrigala*, but also contributes to a broader understanding of its role in sustainable aquaculture. The current findings provide a foundation for optimizing breeding protocols not just for *C. mrigala*, but also for other similar species within the Cyprinidae family, offering insights into their aquaculture viability.

## Figures and Tables

**Figure 1 animals-14-02799-f001:**
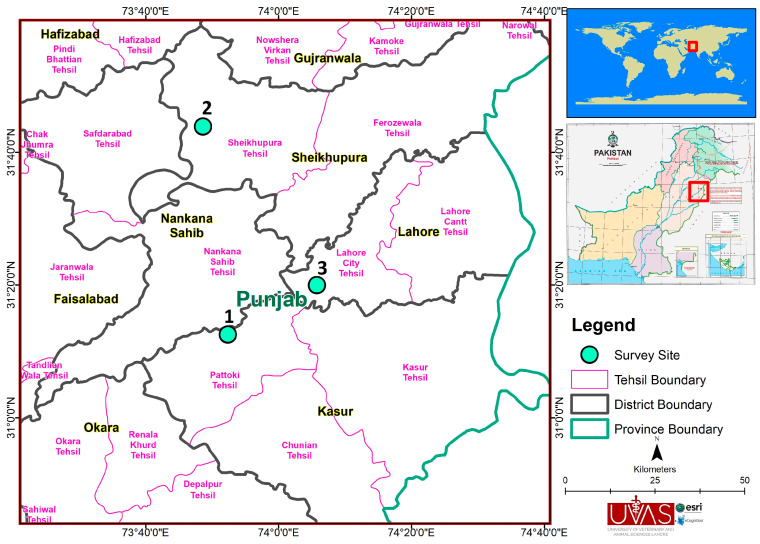
Map highlighting the three study sites.

**Figure 2 animals-14-02799-f002:**
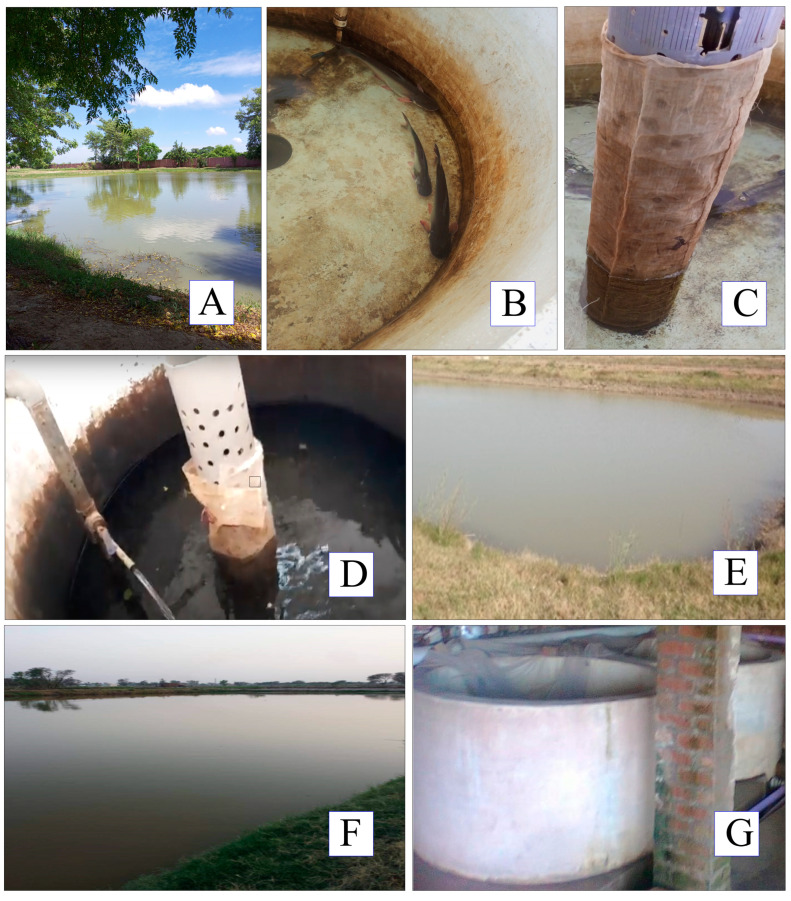
Pictorial view of various embryonic development setups for freshwater carp *Cirrhinus mrigala* (Location 1: (**A**–**C**)), (Location 2: (**D**,**E**)), (Location 3: (**F**,**G**)).

**Figure 3 animals-14-02799-f003:**
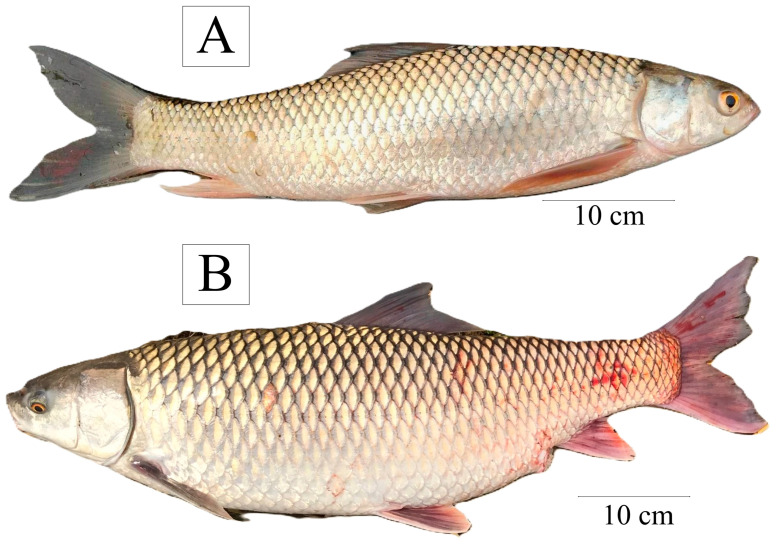
Pictorial view of *Cirrhinus mrigala*, (**A**) male and (**B**) female.

**Figure 4 animals-14-02799-f004:**
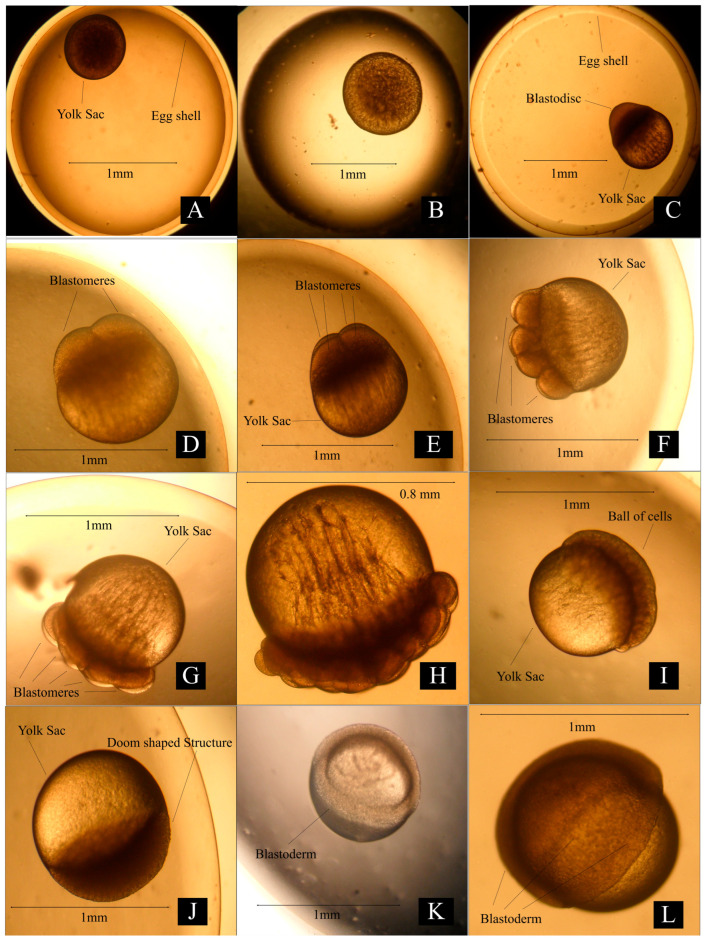
Pictorial view of various embryonic development stages in freshwater carp *Cirrhinus mrigala*. (**A**—unfertilized egg, **B**—fertilized egg, **C**—blastodisc, **D**—2 cell, **E**—4 cell, **F**—8 cell, **G**—16 cell, **H**—32 cell, **I**—early morula, **J**—Late morula, **K**—early Gastrula, **L**—Late gastrula).

**Figure 5 animals-14-02799-f005:**
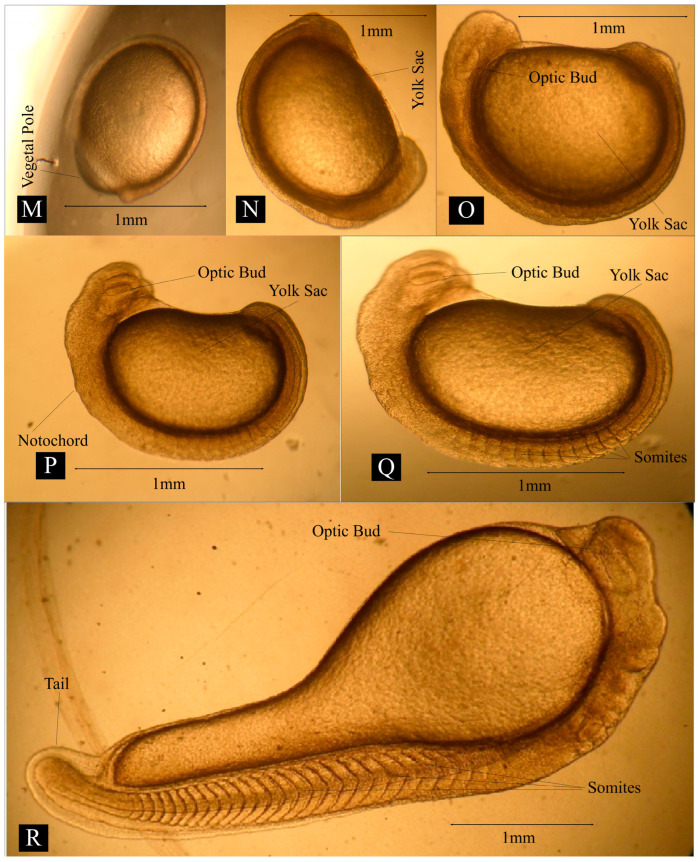
Pictorial view of the optic bud and somites development in freshwater carp *Cirrhinus mrigala*. (**M**,**N**—Early Neurula, **O**—Late Neurula, **P**—Optic Bud formation, **Q**—11–12 somite formation, **R**—25 Somite stage).

**Figure 6 animals-14-02799-f006:**
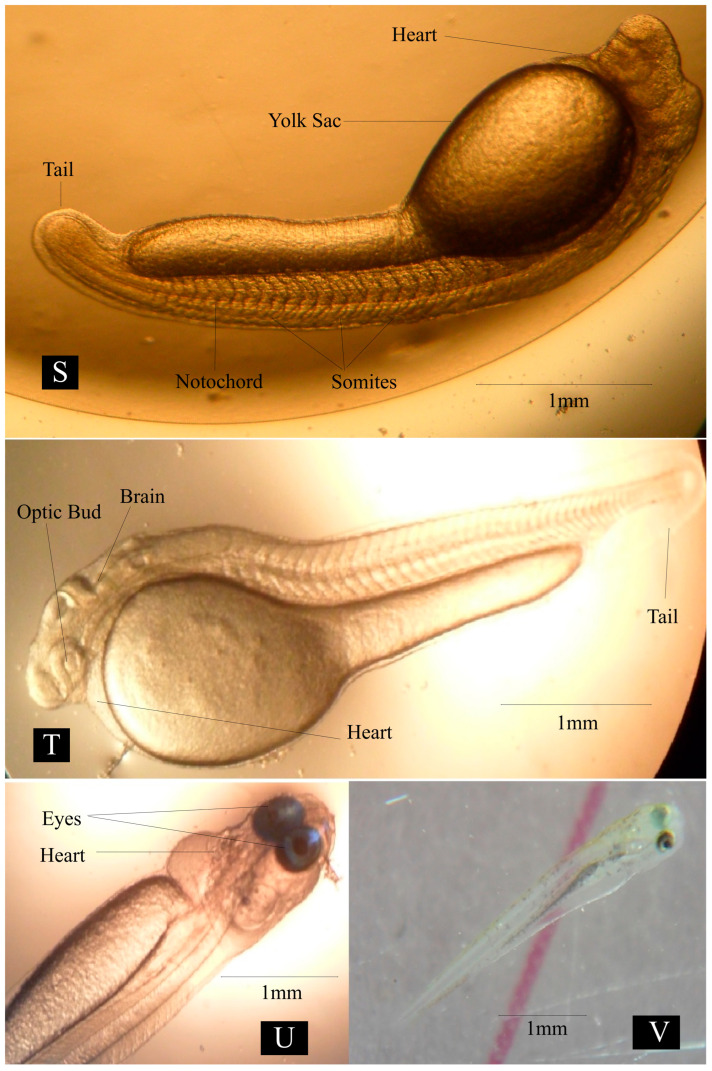
Pictorial view of various organ development stages in freshwater carp *Cirrhinus mrigala*. (**S**—Beginning hatching stage, **T**—Hatching, **U**—Mouth formation, **V**—Swim Bladder formation).

**Table 1 animals-14-02799-t001:** Fish induced spawning and embryonic development setup.

	Description	Location 1	Location 2	Location 3
1	Tank shape	Circular	Circular	Circular
2	Tank type	Cemented	Cemented	Cemented
3	Mesh type	PVC + Nylon Mesh	PVC + Nylon Mesh	PVC + Nylon Mesh
4	Tank height	3.7 feet	4.0 feet	3.5 feet
5	Water inlet diameter (bottom)	0.5 inch	0.75 inch	0.75 inch
6	Water inlet diameter (surface)	0.75 inch	0.75 inch	0.5 inch
7	Volume of water	1208 L	1120 L	1350 L
8	Outlet size (Diameter)	2.5 inch	3 inch	2.5 inch

**Table 2 animals-14-02799-t002:** Sex identification characteristics in *Cirrhinus mrigala* used in the present study.

Character	Male	Female
Size	Smaller than females.	Larger than males.
Body color	Silver gray and bright.	Dark gray back. Silvery appearance on sides and belly.
Body shape	Body bilaterally symmetrical and streamlined, slim.	Dumb bell shaped.
Pectoral fin	Rough to touch.	Smooth to touch.
Vent color and condition	Pink colored vent. Oozes out milt (milky white) upon pressing.	Vent color is darker than male fish. Oozes out eggs (brownish color) upon pressing.

**Table 3 animals-14-02799-t003:** Mean values of physicochemical parameters recorded during trials.

Parameter	Stage Comparison	Mean ± SD	*p*-Value
Temperature (°C)	Before spawning vs. after spawning vs. after hatching.	28.33 ± 1.53, 28.30 ± 0.98, 28.37 ± 1.48	0.87
Dissolved Oxygen (mg/L)	Before spawning vs. after spawning vs. after hatching.	5.29 ± 0.46, 4.56 ± 0.17, 4.15 ± 0.18	<0.01
Electrical Conductivity (mS/cm)	Before spawning vs. after spawning vs. after hatching.	2.04 ± 0.08, 2.08 ± 0.06, 2.02 ± 0.06	0.62
pH	Before spawning vs. after spawning vs. after hatching.	7.33 ± 0.21, 7.47 ± 0.15, 7.33 ± 0.21	0.34
Nitrate (µg/L)	Before spawning vs. after spawning vs. after hatching.	2.07 ± 0.29, 6.25 ± 0.39, 3.50 ± 0.40	<0.01

**Table 4 animals-14-02799-t004:** Dose calculation for induced spawning in freshwater carp *Cirrhinus mrigala* in three locations.

Location × Sex	Weight of Brooder (kg)	Dose of Ovaprim (mL)
Location 1 × Male	1.73 ^b^ ± 0.06	0.35 ^b^ ± 0.01
Location 1 × Female	2.57 ^a^ ± 0.11	1.03 ^a^ ± 0.04
Location 2 × Male	1.67 ^b^ ± 0.08	0.33 ^b^ ± 0.02
Location 2 × Female	2.52 ^a^ ± 0.08	1.01 ^a^ ± 0.03
Location 3 × Male	1.77 ^b^ ± 0.07	0.35 ^b^ ± 0.01
Location 3 × Female	2.49 ^a^ ± 0.16	1.00 ^a^ ± 0.06
*p*-value	<0.0001	<0.0001

^a–b^ superscripts on different means within the column differ significantly at *p* ≤ 0.05.

**Table 5 animals-14-02799-t005:** Time-lapse imaging and egg diameters of *Cirrhinus mrigala*.

Stage Phase	Location 1		Location 2		Location 3	
	Time after Fertilization (min)	Egg Diameter (mm)	Time after Fertilization (min)	Egg Diameter (mm)	Time after Fertilization (min)	Egg Diameter (mm)
Unfertilized eggs	0	2.1	0	2.12	0	2.13
Fertilized eggs	0	2.7	0	2.68	0	2.72
Two-cell stage	30	3.8	26	3.7	31	3.8
Four-cell stage	38	3.9	36	3.9	39	3.8
Eight-cell stage	49	4.1	46	4.3	48	4.5
Sixteen-cell stage	57	4.41	55	4.4	60	4.5
Thirty-two-cell stage	70	4.43	72	4.42	73	4.51
Early morula stage	97	4.47	99	4.45	97	4.51
Mid-morula stage	131	4.49	134	4.47	130	4.51
Late morula stage	177	4.5	179	4.49	180	4.52
Early gastrula stage	239	4.51	242	4.5	246	4.52
Mid-gastrula stage	304	4.53	298	4.54	314	4.52
Late gastrula stage	356	4.54	352	4.54	359	4.52
Early neurula stage	490	4.54	501	4.55	492	4.53
Late neurula stage	566	4.56	553	4.55	567	4.55
Organogenesis	735	4.58	725	4.55	746	4.57
Beginning hatching stage	954	4.58	940	4.55	948	4.57
Just before hatching	1022	4.58	1028	4.55	1036	4.57

**Table 6 animals-14-02799-t006:** Description of embryonic development stages in freshwater carp *Cirrhinus mrigala*.

Figure No.	Stage	Characteristics
Figure 4A	Unfertilized eggs	Unfertilized eggs were circular and their sizes ranged from 2.1 mm to 2.13 mm with circular yolk sacs inside.
Figure 4B	Fertilized eggs	Spherical. Demersal. Non-adhesive. A red spot on one pole.
Figure 4C	Blastodisc stage	Blastodisc formation immediately after fertilization.
Figure 4D	Two-cell stage	The two blastomeres were highly rounded just after the cleavage but were comparatively flat just before the second cleavage.
Figure 4E	Four-cell stage	The second cleavage furrow developed on the two blastomeres at a right angle to the first cleavage plane. It deepened until each blastomere is divided into two blastomeres of the same size.
Figure 4F	Eight-cell stage	The third cleavage plane was parallel to the first and divided the four blastomeres into eight blastomeres.
Figure 4G	Sixteen-cell stage	The fourth cleavage divided the eight cells into sixteen cells.
Figure 4H	Thirty-two-cell stage	The fifth cleavage was reported to continue to occur meridionally at least through the 32-cell stage.
Figure 4I	Early morula stage	The cell number increased and the cells were more tightly packed and a ball of cells was only identifiable upon the surface of the yolk.
Figure 4J	Late morula stage	Appearance of dome shaped structure.
Figure 4K	Early gastrula stage	The blastoderm began to expand (epiboly, about 1/4 of the yolk sphere) over the surface of the yolk sphere. It was difficult to recognize the boundaries of the flattened marginal cells.
Figure 4L	Late gastrula stage	The blastoderm covered 3/4 of the yolk sphere, and the embryonic shield (body) became more clearly visible as a narrow streak.
Figure 5M,N	Early neurula stage	The yolk sphere was nearly covered by the thin blastoderm leaving a small area around the vegetal pole (yolk plug) exposed. A beak-like mass of cells was visible in front of the head.
Figure 5O	Late neurula stage	The brain and nerve cord in the arrow-shaped embryonic body co-developed as a solid rod of cells. The beak-like cell mass was still present.
Figure 5P	Optic bud formation	Notochord became visible, auditory and optic vessels became more developed.
Figure 5Q	11–12 somite formation	Around 11–12 somites were visible.
Figure 5R	25 somite formation	A total of 25 somites were visible.
Figure 6S	Beginning hatching stage	The embryo started to move in the whole egg’s peripheral space. Blood circulation was started.
Figure 6T	Hatching	Newly hatched larvae with slow tail movement were continuously beating the eggshell by the caudal region.
Figure 6U	Mouth formation	Start of feeding (endo-exogenous nutrition).
Figure 6V	Swim bladder formulation	The larvae began to fill the posterior chamber of the swim bladder.

## Data Availability

Data are contained within the article and Appendix A.

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
