# Peer review of "Captive Breeding and Early Developmental Dynamics of Cirrhinus mrigala: Implications for Sustainable Seed Production"

_animals, 2024, doi:10.3390/ani14192799_

Round 1

Reviewer 1 Report

Comments and Suggestions for Authors

All the comments and suggestions are included in the attached word file

Comments on the Quality of English Language

Minor editing in the English can be done, to improve the text.

Author Response

Reviewer No. 1

Review Summary

The manuscript titled “Captive breeding, embryonic and larval development of economically important freshwater carp Cirrhinus mrigala” discuss about the ontogeny and early development of the freshwater carp Cirrhinus mrigala. The aforementioned species is considered an important species for Asian aquaculture, especially those from Pakistan and India. The manuscript describes the early developmental stages from fertilization to early hatching of this species, using macroscopically examination through images. The research can be very helpful for the biology of this species, since it is not described elsewhere. On the other hand, the research is basal, since the state-of-the-art research nowadays, requires multidisciplinary approaches.  Therefore, this manuscript can be fundamental, to furtherly research and examine the candidate species.

Moreover, the manuscript describes practices that are very-well aligned for aquaculture purposes, though it is not very clear why Animals

introduction of the manuscript L81- Considering the aquaculture importance of Cirrhinus mrigala, solid experimentation about early life history is very crucial for the optimization of mass scale seed production, propagation, and management. The  present  study  aimed  to  explore  and deliver a detailed report on the embryonic and larval development of Cirrhinus mrigala to facilitate its successful production and management whereas the aim of the Animals is to publish original research articles, reviews and communications that offer substantial new insight into any field of study that involves animals, including zoology, ethnozoology, animal science, animal ethics and animal welfare that will furtherly assist on the  future research  for  this  finfish  species.  The manuscript can be reconsidered for publication after major revisions, followed below:

Answer: Dear reviewer thanks you very much for your valuable feedback for improvement of this manuscript. We have revised and updated the manuscript as per your kind suggestions. As far as your concern about submission of this article to Animals journal, we want to focus your attention on the fact that Indian Major Carps (Cirrhinus mrigala, Labeo rohita and Catla Catla) are widely cultured species in Pakistan and India region. These fish species are mostly considered as an experimental animal for scientific studies to explore knowledge about growth, reproduction, physiology and the impact of environment. These fish species are native and easily available, so ichthyologist always prefer to undertake these species for scientific experimentation. Cirrhinus mrigala population in Pakistan is also heavily impacted by habitat degradation and overexploitation (Mohsin et al., 2021). So, its need of time to initiate solid and vast experimentation for management and conservation of this aquatic animal.

Introduction

Commen 2: L38: The word Ovaprim is the name of the compound used for induced ovulation. Is not considered a keyword that describes biology and ontogeny. Therefore, it is suggested to be removed and add the keywords Ontogeny/Anatomy since these are more descriptive for this manuscript.

Answer: Dear reviewer we welcome your key comment. Changes have been incorporated as per your kind suggestion.

Commen 3: L45-48. The manuscript mentions that the selected species is very important for the Indian subcontinent, however there is no reasons mentioned. It is suggested to describe the reasons for it, and why the species should be taken into thorough examination for research. The same goes for the embryonic development.

L54-58:  Since the selected species is important,  there  should  be  some  data  and numbers,  that strengthen this statement.

Answer: Dear reviewer the introduction section is now revised and updated as per your instructions. The incorporated changes have been red highlighted in introduction section.

Commen 4: Materials and Methods

Since there are three locations in which the experimentation took place, a map of Pakistan with all the three locations should be given as Figure 1.

Answer: A map highlighting study sites has been added in under heading research location.

Commen 5: L101-102: The physico-chemical parameters were regularly recorded during these trials. Give the average value of all the parameters that were examined. Since the manuscript will set up the ground for mass production for aquaculture purposes, these parameters are crucial for someone who would like to replicate the experiments.

Answer: Dear reviewer It has now been added in results section of revised file as per your recommendation.

Commen 6: L104. Since there are morphological descriptions of male and female, high quality images of the sexes should be given.

Answer: The images of both male and female broodstock has been added in revised manuscript (Figure 2).

Commen 7: L125-132: Since there is incubation for the eggs, a more detailed description of the set up should be given. Apart from the vague information mentioned here, more details should be provided. Some information that should be  included  are  (number  of  eggs/incubator,  size and  volume  of incubator, water temperature, water salinity, nitrate, nitrite concentration, oxygen saturation etc.).

Answer: The requested changes have been incorporated and manuscript has now been updated and highlighted as suggested by the worthy reviewer.

Commen 8: Results: There is no text in the result section. Although the data are in tables and figures, no text that describes the provided data is given. The given data should be descriptive and give additional value to the text that describes the results that were produced from the research.

Answer: Dear reviewer, thank you so much for critical but insightful review. We have addressed your comment. Description of tables and figures in the result section is now revised and updated.

Commen 9: L142-152: Data given to the table 2, is not well aligned, therefore is not very understandable to the reading audition.  Also, some statistics are  mentioned,  however  no  information  on  what statistical analysis was  conducted  is  given.  Therefore, a separated section  on  Material  and  Methods should be added, on what statistical analyses took place.

Answer: Dear reviewer thank you for your genuine comment. Data analysis heading with statistical design is added in material and methods section.

Commen 10: L155-156: Since the morphological description of the early developmental stages is the concept of this manuscript, the table and the followed images, should be more descriptive with details that show the differences among the stages. Below, there are some changes that should be included in the manuscript, to assist the reading audience for better understanding and following up.

  1. a) Indicators, marks and arrows should be included in the photos that have details, based on the developmental stage (e.g., blastodisc formation,  meiotic  spindle,  micropyle,  heart  and  gill rudiment, somites, mouth gap, swim bladder, etc).
  2. b) Scale bars must be included in the photos, to assist the reading audience on the scaling of each developmental stage.
  3. c) Images with better quality and better alignment of the specimen is mandatory. For example, in the photos  1F-1H,  it  is  not  distinguishable  to  understand  the different  stages,  since  the direction/alignment of the examined specimen is not well. Therefore, it is suggested to replace all the photos with better quality ones, were the morphological differences between the developmental stages is easily distinguishable and clarified.

Answer:

Dear reviewer all the suggested changes have been incorporated as per your kind recommendation.

Morphological data is highlighted by specific indicator marks, arrows and detailed description.

Moreover, scale bars have been added in the images to assist the reading audience.

Images have been now refined. We have tried our best to adjust the image quality. Marks and indicators have been applied for better view and understanding of these images.

Discussion

Most of the text in this section, should be included in the result section, since it describes the developmental stages. The whole section should be re-written, based on previous studies, either on the same species and/or other Cypriniformes. Also, a paragraph could be included, to justify the reasons this manuscript should be published in Animals journal (based on its Aims & Scope) and not in other MDPI,s Journal that is specified for aquaculture purposes (i.e., Aquaculture Journal).  Finally, the  conclusions  should  summarize  the  results  of  the  manuscript  (and  not summarize the methods), and how they could  set-up the ground for future studies. Conclusion should be the take-home-message, which is the information that the audience should remember after reading the manuscript.

Answer:

We welcome your solid input. Both discussion and conclusion sections are now revised and updated as per your kind suggestion.

Reviewer 2 Report

Comments and Suggestions for Authors

The authors simply observed and described the embryos development of Cirrhinus mrigala. Before being accepted as a publication in the Journal, the authors should improve the manuscript as below,

1.The authors should comparatively analyze the embryos development under different conditions, eg., temperature, Dissolved Oxygen DO....

2.All the picture should be trimmed and add the scale bars.

Author Response

Reviewer No. 2

Comments 1

The authors simply observed and described the embryos development of Cirrhinus mrigala. Before being accepted as a publication in the Journal, the authors should improve the manuscript as below,

Dear reviewer, we are highly grateful to you for your impactful feedback. We will try our best to satisfy your comments.

Answer:

Dear reviewer, we are highly grateful to you for your impactful feedback. We will try our best to satisfy your comments.

Comments 2

The authors should comparatively analyze the embryos development under different conditions, eg., temperature, Dissolved Oxygen DO.

Answer:

Dear reviewer we welcome your valuable suggestion.

We have already published article on “Impact of Temperature Variations on Breeding

Behavior of Cirrhinus mrigala during Induced Spawning” (Khan et al., 2022) in Pakistan Journal of Zoology.

We will go for more solid experimentation on this fish regarding Effects of other parameters like Dissolved Oxygen on its embryonic and larval development depending on the available funds.

Comments 4

All the picture should be trimmed and add the scale bars.

Answer:

All the images of morphological stages have been revised and updated already by the kind suggestions of worthy reviewer 1.

Morphological data is highlighted by specific indicator marks, arrows and detailed description.

Moreover, scale bars have been added in the images to assist the reading audience.

Images have been now refined. We have tried our best to adjust the image quality. Marks and indicators have been applied for better view and understanding of these images.

Reviewer 3 Report

Comments and Suggestions for Authors

General comments

This is a descriptive paper on the development of freshwater Carp eggs and therefore the title should be changed accordingly. The paper would be improved by completely rewriting the results section and providing additional scientific data. For instance I do not see any data on fertilisation rates hatching rates, sperm mobility or even data on sperm itself. The description of the development stages of the eggs is adequate but the paper can be much improved.

Specific comments

I suggest using different term to breeding as this intimates the bringing together of male and female fishes. In this case this is artificial as both eggs and sperm are stripped from females and males then fertilised in vitro. Perhaps Captives Seed Production

Page 1 line 22: not a rate  : please use dose

Page 2 line 70: artificial seed production

Page 2 line 81: solid descriptive scientific evidence?

Page 3 line 111:” at a dosage of”

Page 4 line 134: time-lapse photography

Page 4 The results of this section is completely lacking in text:  This is not acceptable as the results sections should consist of text, tables and figures.

table 2: is not displaying properly in my manuscript with the dose of Oviprim  displaced left.

Table 3: these mean values should show + standard deviations!

Figure 1 requires a scale bar on all of the pictures!

Discussion

page 10 line  201: the authors should also consider the effective of incubation temperature on the developmental times.

Comments on the Quality of English Language

Minor editing of English language required.

Author Response

Reviewer No. 3

General comments

Comments 1: This is a descriptive paper on the development of freshwater Carp eggs and therefore the title should be changed accordingly. The paper would be improved by completely rewriting the results section and providing additional  scientific  data.  For instance I do not see  any  data  on  fertilization  rates  hatching  rates,  sperm  mobility  or  even  data  on  sperm  itself. The description  of  the  development stages of the eggs is adequate but the paper can be much improved.

Answer:

Dear reviewer we have very heartfelt gratitude towards your very valuable feedback for improvement of this manuscript. We have done efforts to address your genuine comments.

Manuscript title has been revised and highlighted.

Sections like, material and methods, results and discussion have been revised and updated and highlighted already as per kind suggestions of reviewer 1.

Specific comments 2:

I suggest using different term to breed as this intimates the bringing together of male and female fishes. In this case this is artificial as both eggs and sperm are stripped from females and males then fertilized in vitro. Perhaps Captives Seed Production

Answer:

Alright its replaced and highlighted by term Artificial/induced breeding in revised manuscript as per your kind suggestion.

Comments 3: Page 1 line 22: not a rate: please use dose

Answer:

Alright term dose is incorporated and highlighted in revised manuscript.

Comments 4: Page 2 line 70: artificial seed production

Answer:

Suggested term is incorporated and highlighted.

Comments 5: Page 2 line 81: solid descriptive scientific evidence

Answer:

Revised and updated as per your kind suggestion.

Comments 6: Page 3 line 111:” at a dosage of”

Answer:

Suggested words added and highlighted.

Comments 7: Page 4 line 134: time-lapse photography

Answer:

Added and updated.

Comments 8: Page 4 The results of this section is completely lacking in text: This is not acceptable as the results sections should consist of text, tables and figures.

Answer:

Dear reviewer the result section is completely revised and improved already as per kind instruction of reviewer 1.

Comments 9: Table 2: is not displaying properly in my manuscript with the dose of Ovaprim displaced left.

Answer:

Table is now revised and updated and designated as table 4 in revised manuscript.

Comments 10: Figure 1 requires a scale bar on all the pictures!

Answer:

Scale bar is added and now figure 1 is divided into figure, 4,5 and 6 for better visualization and understanding.

Comments 11: Discussion: page 10 line 201: the authors should also consider the effective of incubation temperature on the developmental times.

Answer: Dear reviewer we welcome your valuable suggestion.

Discussion section is revised and improved as per kind instruction of reviewer 1.

We have already published article on “Impact of Temperature Variations on Breeding

Behavior of Cirrhinus mrigala during Induced Spawning” (Khan et al., 2022) in Pakistan Journal of Zoology.

We will go for more solid experimentation on this fish regarding Effects of other parameters like Dissolved Oxygen on its embryonic and larval development depending on the available funds.

Comments 12: Comments on the Quality of English Language. Minor editing of English language required.

Answer:

Dear reviewer revised version of manuscript has been improved through extensive English language editing.

Round 2

Reviewer 1 Report

Comments and Suggestions for Authors

The manuscript is reviewed based on the given comments. However, still some issues remain

 Introduction

-The references in L67 don’t follow the right numeric order.

-Also, add some information about the aquaculture of the species (annual production in tons or kilos and commercial value of it in dollars, if there are any such information).

 Material and methods

-L104: Add scale bars to the images for size comparison

-Table 1: Replace commas (,) with full stops (.) in values where is necessary.

Results

-This section still needs full revision. In this section, there are two ingredients. First, an overall description of the experiments should be given, providing the big picture without repeating the experimental details that are described in material and methods. Second, you should present the data in the past tense. In other words, all the figures and tables (apart from the given format) should be written down as text, in order to help the reading audience to fully understand the morphology, biology, anatomy and development of the examined species. Some advice:

1. The results need to be clearly and simply stated, because it is the results that constitute the new knowledge that is contributed to the world.

2. Avoid redundancy. A usual mistake is the repetition of words, which are apparent to the reader audience from the examination of figures and tables

For more assistance, there are plenty manuscripts on the internet with similar studies on other species (e.g. Sparus aurata, Dicentrarchus labrax, Dentex dentex, etc), that assist on writing properly the result section.

-Replace Images 4F-4H, since it seems to be the same developmental stage from different angle and magnification! The developmental differences of these photos are not distinguishable. More accurate photos are required!

-Supplementary files are missing. If there are any, should be uploaded to the platform and also included there input in the text (e.g., Supplementary Table 1, Supplementary Figure 1, etc). Revise the whole manuscript that include the supplementary files also (if any).

Author Response

We are highly grateful to you for your critical but profound feedback that will definitely improve our manuscript. This manuscript off coarse after revision will be a nice piece of work for scientific readers and farmers community as well. We have tried our best to address all the shortcomings in the revised manuscript. Hopefully, it will serve the purpose.

The change (-s) suggested by the reviewer are highlighted with red color fonts for better understanding. The manuscript can be reconsidered for publication after major revisions, followed below:

Reviewers Comments:

Introduction

-The references in L67 don’t follow the right numeric order.

-Also, add some information about the aquaculture of the species (annual production in tons or kilos and commercial value of it in dollars, if there are any such information).

Answer:

Changes have been incorporated and highlighted as per your kind instructions.

Reviewers Comments:

Material and methods

-L104: Add scale bars to the images for size comparison

-Table 1: Replace commas (,) with full stops (.) in values where is necessary.

Answer:

Scale bar has been added in the size comparison chart of male and female fish in the revised manuscript.

Table 1 has been modified as per your kind suggestion.

Reviewers Comments:

Results

-This section still needs full revision. In this section, there are two ingredients. First, an overall description of the experiments should be given, providing the big picture without repeating the experimental details that are described in material and methods. Second, you should present the data in the past tense. In other words, all the figures and tables (apart from the given format) should be written down as text, in order to help the reading audience to fully understand the morphology, biology, anatomy and development of the examined species. Some advice:

1. The results need to be clearly and simply stated, because it is the results that constitute the new knowledge that is contributed to the world.

2. Avoid redundancy. A usual mistake is the repetition of words, which are apparent to the reader audience from the examination of figures and tables

For more assistance, there are plenty manuscripts on the internet with similar studies on other species (e.g. Sparus aurata, Dicentrarchus labrax, Dentex dentex, etc), that assist on writing properly the result section.

-Replace Images 4F-4H, since it seems to be the same developmental stage from different angle and magnification! The developmental differences of these photos are not distinguishable. More accurate photos are required!

-Supplementary files are missing. If there are any, should be uploaded to the platform and also included there input in the text (e.g., Supplementary Table 1, Supplementary Figure 1, etc). Revise the whole manuscript that include the supplementary files also (if any)

Answer:

Dear reviewer we welcome your suggestions for improvement of result section.

  1. Result section has been revised with appropriate text for its better presentation.

  2. The detailed description of results has removed the chance of words repetition.

  3. Dear reviewer thank you for your comment. Actually we have these pictures only regarding these typical developmental stages. We have done some efforts to present it clearly. So please accept these in current form.

  4. Dear reviewer supplementary file regarding physicochemical parameters of water during induced spawning has been provided. All the remaining data generated during this study is already presented in the manuscript.

Reviewer 2 Report

Comments and Suggestions for Authors

The manuscript has been improved much,but its contribution to the readership  is still limited.

Comments on the Quality of English Language

N.A.

Author Response

We appreciate your comments on our MS. The manuscript has been greatly improved, and much information has been added, even in the form of supplementary Tables.

Reviewer 3 Report

Comments and Suggestions for Authors

Thank you for editing 

Author Response

Thank you very much for your constructive comments regarding our MS, thanks to which our article has gained in quality.

Round 3

Reviewer 1 Report

Comments and Suggestions for Authors

The manuscript is significantly improved from the last version. Figure 4 has to be replaced, since the third row of images is not clearly seen.

Author Response

Response to reviews

Comments 1: The manuscript is significantly improved from the last version.

Response 1: Thank you very much for your time and comments which helped us improve our MS.

Comments 2:  Figure 4 has to be replaced, since the third row of images is not clearly seen.

Response 2: Thank you for your feedback regarding Figure 4. Upon reviewing the figure, we believe the images provide necessary details, particularly in illustrating key aspects of the data. We selected the highest quality photos available for the manuscript, but capturing clear images was challenging due to the size of the developing eggs, which were about 1 mm in diameter. As the eggs matured, they became less transparent, making it increasingly difficult to achieve ideal clarity. Nevertheless, the most important details remain visible and should be clear to the reader.
